# Nutrition-Related Information on Alcoholic Beverages in Victoria, Australia, 2021

**DOI:** 10.3390/ijerph19084609

**Published:** 2022-04-11

**Authors:** Katerina Penelope Barons, Davina Mann, Liliana Orellana, Mia Miller, Simone Pettigrew, Gary Sacks

**Affiliations:** 1School of Health and Social Development, Deakin University, Burwood, VIC 3125, Australia; kbarons@deakin.edu.au; 2Global Obesity Centre, Institute for Health Transformation, Deakin University, Geelong, VIC 3220, Australia; d.mann@deakin.edu.au; 3Biostatistics Unit, Deakin University, Geelong, VIC 3220, Australia; l.orellana@deakin.edu.au; 4The George Institute for Global Health, University of New South Wales, Sydney, NSW 2052, Australia; mmiller@georgeinstitute.org.au (M.M.); spettigrew@georgeinstitute.org.au (S.P.); 5The Menzies School of Health Research, Charles Darwin University, Darwin, NT 0811, Australia

**Keywords:** alcohol, nutrition, labelling, energy, nutrients, obesity

## Abstract

Alcoholic beverages sold in Australia are largely exempt from requirements to display nutrition information on packages, unlike other food and beverages. However, alcoholic beverage manufacturers can provide nutrition-related information voluntarily. This study aimed to investigate the prevalence of nutrition-related information on packaged alcoholic beverages in Australia. An in-store audit of the largest alcohol retailer in Melbourne, Australia was conducted in July 2021. A systematic sampling method was used to assess the presence and format of nutrition information on 850 alcoholic beverages across 5 alcohol categories (wine (*n* = 200), beer (*n =* 200), spirits (*n =* 200), ready-to-drink beverages (*n =* 140) and ciders (*n =* 110)). Most products (*n =* 682, 80.2%) did not present nutrition-related information. Where information was presented (*n =* 168), it was most frequently on ready-to-drink beverages (*n =* 81, 57.9%) and least frequently on spirits (*n =* 9, 4.5%) and wines (*n =* 9, 4.5%). Nutrition information was most frequently in the format of a nutrition information panel (*n =* 150, 89.3%) and approximately half of labelled beverages (*n =* 86, 51.2%) included a nutrition content claim (e.g., ‘low in carbs’). Given limited voluntary implementation of nutrition labelling on alcoholic beverages in Australia and the substantial contribution of alcoholic beverages to energy intake, consideration of mandatory nutrition labelling, in a standardised format designed to maximise public health benefit, on alcoholic beverages is warranted.

## 1. Introduction

Alcoholic beverages are one of the key contributors to the burden of disease globally and in Australia [1,2]. Indeed, consumption of alcoholic beverages has been associated with a range of negative health and social consequences including increased risk of several types of cancer, chronic liver disease and family violence [3,4,5,6]. In addition, in Australia, alcoholic beverages are one of the greatest contributors to energy intake, contributing 4.8% of average adult daily energy intake at the population level [3]. Excess energy intake has driven the high prevalence of obesity rates in Australia in recent decades [2,7].

The nutrient and energy content of alcoholic beverages varies widely by category (e.g., wine, beer, spirits) and within categories. Alcohol is relatively high in energy content, with one standard drink of alcohol (defined in Australia as any drink containing 10 g of alcohol) containing at least 290 kJ from alcohol content alone [4]. Ready-to-drink alcoholic beverages (RTDs, or “alcopops”), in which alcohol has been pre-mixed with a non-alcoholic mixer such as sugary soft drinks, tend to be highest in energy content [8].

Nutrition labelling can help individuals understand the nutritional content of different foods and beverages. Globally, nutrition labelling is regulated by a mix of international standards (including the Codex Alimentarius [9]), government regulations and guidelines, and voluntary industry practices [10]. On-product nutrition content labelling can be presented as part of a Nutrition Information Panel (NIP) (typically on the back of product packaging), or in other formats, including a front-of-pack (FOP) label, for example the Health Star Rating or Nutri-Score labelling systems, or in a statement on the back of the beverage, for example “this product contains 100 g of sugar”. Nutrition-related information may also be presented in the form of a nutrition content claim or health claim [10,11]. In Australia, nutrition content claims, also referred to as nutrition claims, are statements regarding the food or beverage’s nutrient content (e.g., “low in sugar”), typically based on minimum criteria [12]. Health claims describe how a food or beverage’s nutritional content may benefit health, for example “supports a positive change in gut health” [12]. Under national regulations, alcoholic beverages sold within Australia can only present nutrition content claims regarding energy, carbohydrate/sugars, or gluten content, and cannot present health claims [12]. When an approved nutrition content claim is presented, the product must also display an NIP describing the energy, protein, fat, saturated fat, carbohydrate, sugars, and sodium contents of the product. The presence of nutrient content information, such as “no sugar” on the FOP label does not require an NIP to also be presented. Under Australian regulations, alcoholic beverages that do not make a nutrition content claim are not required by law to provide any nutrition content information, as is the case in other countries, including Canada [12,13]. Other jurisdictions have different requirements, for example countries belonging to the Eurasian Economic Union mandate presentation of the beverage’s energy content on all alcohol labels [14]. The global alcohol industry has committed to improving alcohol nutritional labelling through more consistent and user-friendly presentation of nutritional content on alcoholic beverages [14,15,16]. The recently released Australian National Preventive Health Strategy 2021–2030 includes a recommendation that energy content information should be presented on all packaged alcoholic products. The Australian government is in the process of reviewing current regulations around nutrition-related information labelling of alcoholic beverages, including exploring options for presenting energy content information on alcohol labels and the formats in which this information should be presented [17,18].

There is evidence that nutrition labels on non-alcoholic food and beverage products can encourage healthier choices, thus helping to improve dietary patterns [19,20,21]. For example, a 2020 systematic review and meta-analysis of 14 studies found FOP labels that presented nutrient information significantly reduced the average energy, sugar, and sodium content of purchased food items [19]. In addition, the presence of nutrition content information labelling in the form of NIPs tends to increase the number of people selecting a healthier food product over a like-for-like less healthy food product [19,20,21]. However, there is some evidence that the presence of nutrition information has been found to create a ‘health halo’ effect, in which people perceive a product to be healthier than it actually is [22,23]. As a consequence, although nutrition labelling can encourage purchases of healthier products, in some cases, it may also lead to increased and excessive consumption of less healthy products.

Limited research has explored the impact of the provision of nutrition information on alcoholic beverages specifically. The few existing studies in this area indicate that nutrition information can help deter consumers from purchasing alcoholic beverages, such as RTDs, that contain high amounts of ‘risk nutrients’ (such as sugar) but may be less effective in deterring purchases of other alcoholic beverages, like wines, with relatively lower amounts of risk nutrients [24,25,26]. As is the case with food products, research also indicates that nutrition labels can cause a halo effect with alcohol products [27]. One study indicated that alcohol products marketed as ‘better-for-you’ created an illusion of healthiness, despite the potential adverse impacts of the products on population health [27]. In general, there is evidence that the public typically underestimates the energy and nutrient content of alcoholic beverages [28]. Nutrition content claims, like ‘low-kilojoule’, may further hinder consumers’ ability to understand the health impact and nutrient contents of alcoholic beverages [28]. However, the provision of nutrition information on alcohol labels can increase people’s understanding of the nutrient content of alcoholic beverages and encourage the selection of lower-energy alcoholic drinks [29].

To date, only a small number of studies globally have investigated the prevalence of nutrition information on alcoholic beverages. A 2013 literature review assessing the Australian market noted that most alcoholic beverage labels did not display nutrition information. Two studies in Europe, one in 2018 and another in 2016, also noted that alcoholic beverage labels seldom present nutrition information in any form [30,31]. The format (e.g., NIP or nutrition content claims) in which nutrition content information on alcoholic beverages is currently presented, in both international contexts and within Australia, has not been explored in detail. Further, there is little available evidence regarding the differences in prevalence of nutrition-related information by category of alcohol. This therefore limits an understanding of the current use of nutrition-related information labelling in the alcohol market. This study aimed to explore the prevalence and format of nutrition-related information on alcoholic beverages in Australia, including variations across alcohol categories.

## 2. Materials and Methods

### 2.1. Study Design

An in-store audit was conducted at the largest store in Victoria of an alcohol retail chain that is the market leader in Australia, with a market share of over 60% of the alcohol retail market [32]. A stratified sample was used where alcoholic beverages were the sampling unit. Strata included beer, wine, spirits, RTDs and cider. These strata are consistent with the categorisation of alcohol by the Australian government [33], the categories with the greatest proportion of consumption within Australia [34] and previous categorisations applied in the alcohol audit literature [35,36,37].

In total, 850 beverages were audited, with the chosen sample size informed by the results of: (1) a preliminary online audit of alcoholic beverages that were available on the website of the largest Australian alcohol retailer (June 2021); and (2) the sample sizes of previous studies that had explored the prevalence and format of food labels [38,39,40]. Approximately 50% of all ciders (*n* = 110) and RTDs (*n =* 140) were included, based on the relatively high prevalence of nutrition information in these categories observed in the preliminary online audit. Approximately 25% of all beers (*n =* 200) were included, with the number of wines and spirits matched to the number of beers (*n =* 200). The details of the number of beverages available for purchase online and the number of beverages included in the study sample are presented in Table 1.

Within each alcohol category, a systematic sampling technique was used. Every second beverage, moving right to left and from the top row to the bottom row of the shelves for each alcohol category were audited. The first audited shelf for each sample category was randomly selected. Once the first shelf had been selected by the auditor, auditing began from the right side of the top row of this shelf. Within the retailer’s store, beverages were organised along their shelves by type of alcohol and not by manufacturer. Selecting every second beverage ensured that the sample included beverages from most or all shelves of beverages for each category and included a good representation of all manufacturers. Beverage availability in store was limited in comparison to the online store for this retailer. As such, the sample included a higher proportion of beverages that were available in-store compared to those available only online. For spirits and wines, the sampling method resulted in approximately half of the accessible beverages being included in the audit. For beers, ciders and RTDs, the sampling method resulted in most accessible beverages being included in the audit. For some alcohol categories, namely spirits and wines, some higher-priced beverages were contained in a locked cabinet. In these instances, the products on these shelves were not included in the sample as the back of the beverage could not be examined for nutrition-related information labelling. However, as the number of shelves that were inaccessible and therefore not included was minimal, this did not have a significant impact on the selected sample. Sampling continued until the target sample size for each category was achieved. For all alcohol categories, the sample included beverages from all accessible shelves. As such, the retailer’s shelf organisation, for example by type or origin of alcohol, had minimal impact on the sample. Where a beverage was available in different sizes, for example in either a small or large bottle, only one size was selected.

### 2.2. Data Collection and Analysis

Data were recorded on a self-developed audit tool (see Appendix A) which was based on previous literature assessing food labelling formats [29,30,31]. The audit tool facilitated documentation of the presence of any nutrition-related information, including both nutrition content claims (e.g., low in sugar) and nutrient/energy levels. The latter refer to presentation of nutrient content in numeric form (e.g., 0 g sugar), in formats including within an NIP or as a statement on the label. Where nutrition-related information was present, we collected the nutrition component to which it related (e.g., energy, sugars, carbohydrate) and the format in which it was provided i.e., a NIP, FOP label, nutrition content claim or a written statement of the nutrient/energy content on the back or front of the package, for example “this product contains 110 calories”. When information was positioned towards the side of the label, they were noted as back labels. The name of the alcoholic beverage manufacturer was captured to assess provision of nutrition information by different companies.

Data were collected in store over a three-day period in July 2021. Photos of all alcoholic beverages that presented nutrition-related information were captured on a smartphone. Photos were transferred to an electronic copy of the audit spreadsheet (Microsoft Excel [41]) once the manual audit had concluded. The photos were used to verify that the nutrition-related information on labels was correctly documented.

An ethics exemption was provided by Deakin University, as this study did not involve the collection or use of human data. Store manager approval was obtained prior to the commencement of data collection.

To test for the inclusion of nutrition-related information labelling, descriptive statistics including frequency, prevalence and range were calculated for the overall sample and by alcohol category.

To test for differences in presence of nutrition-related information between alcohol categories (wines, beers, spirits, RTDs and ciders), a Chi-squared test was performed. Where statistically significant associations were identified (*p* < 0.05), post hoc pairwise comparisons between alcohol categories with Bonferroni correction to account for multiple comparisons were conducted to minimise risk of false positives. Clustering induced by manufacturers was not considered in the analytical approach as the median number of beverages per manufacturer was 1, with 576 manufacturers represented only once in the sample. A descriptive analysis of the manufacturer effect on labelling is presented for the 17 manufacturers that were represented five or more times in the sample. Stata Statistical Software: Release 16 (StataCorp, College Station, TX, USA) [42] was used for all analyses.

## 3. Results

The prevalence of nutrition-related information by alcohol category are presented in Table 2. Most alcoholic beverages sampled (80.2%) did not present any nutrition-related information. The alcohol category with the highest prevalence of nutrition-related information was RTDs (57.9%) and the lowest were wine (4.5%) and spirits (4.5%). The prevalence of nutrition-related information labels was significantly different across alcohol categories (*p* < 0.001), as calculated by Chi-squared tests and post hoc pairwise comparisons between categories with Bonferroni correction (see Appendix A).

In approximately half of the cases (86/168, 51.2%) where nutrition-related information was present, it was presented in the form of a nutrition content claim (Table 2). In line with existing regulations, no beverages presented a nutrition content claim without presenting an NIP and no health claims were made on any of the sampled alcoholic beverages. RTDs had the highest prevalence of nutrition content claims (30.7%), while no claims were found on spirits.

Where nutrient/energy levels were provided, they were typically presented in the form of a NIP only (80/168, 47.6%), or a NIP and a statement on the back of the beverage (70/168, 41.7%). Only three products (1.8%) had nutrition content information in the form of a FOP label. The format of these FOP labels consisted of the industry-designed percentage daily intake label (*n =* 1) and Facts Up Front label (*n =* 2). These FOP label formats are ‘non-interpretive’ in that they do not provide guidance (e.g., using colours or symbols) as to how to interpret the information.

As shown in Table 3, of the 168 alcoholic beverage products that displayed nutrition content information, all presented the beverage’s energy content (100.0%), 154 (91.6%) presented the beverage’s carbohydrate content and 153 (91.1%) presented the beverage’s sugar content. Where an NIP was used (150, 89.3%), the protein, fat and sodium contents were presented in all cases along with the energy, carbohydrate, and sugar contents. Nutrient content claims that related to energy, carbohydrate and/or sugar content were presented in similar proportions (Table 3). Examples of such claims included “half the calories”, “lower carb” and “low in sugar”.

Of the 850 audited alcoholic beverages, 576 discrete manufacturers were identified. The category with the largest number of manufacturers in the sample was spirits (180 manufacturers from 200 beverages), and the category with the lowest number of manufacturers was RTDs (54 manufacturers from 140 beverages). Of the 576 manufacturers, 17 produced five or more beverages included in this study. Of these 17 manufacturers, three presented nutrition-related information on 90% or more of their beverages and four presented nutrition-related information on 10% or less of their beverages, as shown in Table 4.

## 4. Discussion

The results of this study suggest that most alcoholic beverages sold in Australia do not present any nutrition-related information on their labels. When nutrient levels were presented, it took various formats, including as an NIP, a statement on the back of the beverage, or a non-interpretive FOP label. Very few alcoholic beverage manufacturers provided nutrition-related information universally across their product range, with several manufacturers providing such information on only a small subset of their range.

This study found important differences in the prevalence of nutrition-related information by alcohol category. RTDs presented nutrition-related information most frequently (57.9% of total sampled RTDs) and wines and spirits least frequently (4.5% and 4.5%, respectively). RTDs are most frequently consumed by younger (aged 14–24 years) females [34], and there is evidence from food labelling studies that young females use nutrition labelling information more than any other population when choosing a food or beverage [21,43]. There is also evidence that younger females are more concerned about energy intake than other population groups [21,43]. In contrast, wine and spirits are consumed in higher proportions by older age groups [34], and there is evidence that older population groups are typically less influenced by nutrition information [21]. The selective use of nutrition-related information and the pattern of its provision across categories implies that alcoholic beverage manufacturers are likely using this information as a marketing technique to appeal to different demographic groups, rather than as a mechanism to alert consumers to the health risks associated with alcohol consumption. However, the reasons behind alcoholic beverage manufacturers’ decisions to include or not include nutrition-related information on their products and the impact of nutrition-related information on the alcohol consumption decisions of different population groups warrant further exploration.

The study found that most alcoholic beverages that presented nutrient content information included a nutrition content claim related to the beverage’s energy, carbohydrate, or sugar content. There is considerable evidence to indicate that the presence of a nutrition content claim on food products may lead consumers to believe the product is healthier than it is [44,45,46,47,48]. Although there has only been a limited number of studies investigating the impact of nutrition content claims on alcoholic beverages specifically, evidence indicates that their presence is likely to increase perceptions of the healthiness of alcohol products and can exacerbate existing low levels of understanding of alcohol’s energy contribution and other health risks associated with the consumption of alcohol [28]. As such, there are indications that the presence of nutrition content claims may create a ‘health halo’ effect for alcohol products [27]. While more research is needed in this area, regulations restricting nutrition content claims on alcoholic beverages may be required as part of broader efforts to improve population diets and reduce alcohol-related harms. One potential method to enhance the communication of risks related to alcoholic beverage consumption may be to mandate the display of warning labels that describe potential health and social impacts related to alcohol consumption, for example an increased risk of cancer or chronic liver disease [3,4,5,6]. Warning labels have been found to be an effective tool to raise awareness of such risks and subsequently reduce per capita alcohol use [49]. The way in which different types of information provided on product packaging interact and their likely impact warrants further investigation. Such studies need to consider the strong body of evidence that shows the ways in which the alcohol industry seeks to avoid regulation. In addition, the potential impacts of other strategies to limit risks associated with consumption of alcoholic beverages need to be explored, including the impact of other in-store marking techniques, such as price promotions and in-store signage [27,50,51]. Indeed, the role of information labelling on alcoholic beverages needs to be considered as one part of a series of policy mechanisms (including price regulation and availability) that can be used to develop a comprehensive strategy to curb alcohol consumption [52].

This study has several strengths. To our knowledge, it is the first Australian study to explore the prevalence and format of nutrition labelling on alcoholic beverages. Several factors contributed to increasing the representativeness of the sample across the Australian alcohol market, including auditing a retailer with a high market share, the relatively large sample size, and inclusion of the most popular alcoholic beverage categories and products. Several limitations should be acknowledged. Only using one store for this audit may limit the representativeness of the sample and does not allow for assessment of possible differences between retailers, between stores in different geographic locations, and between online and in-store environments. Future studies should examine the prevalence of nutrition-related information on alcoholic beverages in different contexts and over time, including international comparisons. It is currently unknown how consumers perceive and value nutrition information on alcohol products when it is present on some, but not all alcohol products. There is also limited knowledge of whether alcohol drinkers differ in their understanding and/or response to nutrition information, compared with non-drinkers. These aspects warrant exploration as part of future research. Although the systematic sampling technique used in this study was efficient and was designed to minimise selection bias, sampling was not truly at random and product selection may have been impacted by the positioning in the store and along the aisles. Future studies should consider alternate sampling methods and larger sample sizes. While the study examined the format of nutrition-related information, it was not designed to compare the nutrition content of different products, including within and between categories and by manufacturer. Such studies may be valuable in understanding trends in the market, including developments such as lower carbohydrate products and low alcohol and alcohol-free products. Further, this study did not investigate whether the alcoholic beverages that did or did not present nutrition-related information labels differed in alcohol content. Future studies could investigate if the presence of nutrition-related information labelling is associated with alcohol strength.

## 5. Conclusions

There is a low prevalence of nutrition-related information on packaged alcoholic beverages in Australia, with prevalence differing between alcohol categories. The limited voluntary implementation and selective provision of nutrition-related information suggests that alcoholic beverage manufacturers are likely using nutrition-related information as a marketing tool that may increase alcohol consumption. Given the potential adverse health outcomes that can result from alcohol consumption, options to regulate the use of nutrition-related information labelling as a marketing technique should be explored. The mandatory display of warning labels on alcohol beverages also needs to be considered. The currently largely unregulated labelling of alcoholic beverages reflects a missed opportunity to inform people about the high energy content and poor nutritional value of alcoholic beverages, and to highlight the potential negative health and social impacts that can result from alcohol consumption. Consequently, nutrition-related labelling and warning labels should be used as a tool to alert consumers to the potential negative health outcomes that are associated with alcohol consumption. While further research is needed to understand the most effective format for nutrition-related labelling of alcoholic beverages from a public health perspective, consideration of mandatory nutrition labelling on alcoholic beverages, coupled with restrictions on nutrition content claims, is warranted as part of broader efforts to improve population diets and address alcohol-related harm.

## Figures and Tables

**Table 1 ijerph-19-04609-t001:** Total number of beverages (by category) available online from the alcohol retailer (June 2021) and included in the study sample for the in-store audit (July 2021).

Alcohol Category	Total	Beers	Wines	Spirits	Ready-to-Drink Beverages	Ciders
Number of beverages available online	9902	830	6644	1925	283	220
Number of beverages included in the study sample	850	200	200	200	140	110

**Table 2 ijerph-19-04609-t002:** Frequency and prevalence nutrition-related information presented on alcoholic beverages (by format and alcohol category) in a sample of 850 alcoholic beverages for sale in Victoria, Australia in July 2021.

Information Presented	Total (*n* = 850)	Beers (*n* = 200)	Wines (*n* = 200)	Spirits (*n* = 200)	Ready-to-Drink Beverages (*n* = 140)	Ciders (*n* = 110)
**Nutrition-related information present**	168 (19.8%)	58 (29.0%)	9 (4.5%)	9 (4.5%)	81 (57.9%)	11 (10.0%)
** *Nutrient levels BUT no nutrition content claim* **	82 (9.7%)	31 (15.5%)	1 (0.5%)	9 (4.5%)	38 (27.1%)	3 (2.7%)
** *Both nutrient levels AND a nutrition content claim* **	86 (10.1%)	27 (13.5%)	8 (4.0%)	0 (0.0%)	43 (30.7%)	8 (7.3%)
**Format of nutrient levels (where present)**						
**Nutrition information panel only**	80 (47.6%)	38 (65.6%)	9 (100.0%)	4 (44.4%)	18 (22.2%)	11 (100.0%)
**Nutrition information panel and statement on back of beverage**	70 (41.7%)	8 (13.8%)	0 (0.0%)	2 (22.2%)	60 (74.1%)	0 (0.0%)
**Statement on back of beverage only**	15 (8.9%)	12 (20.7%)	0 (0.0%)	3 (33.3%)	0 (0.0%)	0 (0.0%)
**Statement on back of beverage and front-of-pack label**	1 (0.6%)	0 (0.0%)	0 (0.0%)	0 (0.0%)	1 (1.2%)	0 (0.0%)
**Front-of-pack label only**	2 (1.2%)	0 (0.0%)	0 (0.0%)	0 (0.0%)	2 (2.5%)	0 (0.0%)

**Table 3 ijerph-19-04609-t003:** Frequency and prevalence of nutrition-related information presented on alcoholic beverages (by content type and alcohol category) in a sample of 850 alcoholic beverages available for sale in Victoria, Australia in July 2021.

	Total (*n =* 850)	Beer (*n =* 200)	Wine (*n =* 200)	Spirits (*n =* 200)	Ready-to-Drink Beverages (*n =* 140)	Cider (*n =* 110)
**Total number of products presenting nutrient/energy levels**	168 (19.8%)	58 (29.0%)	9 (4.5%)	9 (4.5%)	81 (57.9%)	11 (10.0%)
- *Energy content*	168 (19.8%)	58 (29.0%)	9 (4.5%)	9 (4.5%)	81 (57.9%)	11 (10.0%)
- *Carbohydrate content*	154 (18.1%)	48 (24.0%)	9 (4.5%)	6 (3.0%)	81 (57.9%)	10 (9.1%)
- *Sugar content*	153 (18.0%)	47 (23.5%)	9 (4.5%)	6 (3.0%)	81 (57.9%)	10 (9.1%)
- *Protein content*	150 (17.6%)	46 (23.0%)	9 (4.5%)	6 (3.0%)	78 (55.7%)	11 (10.0%)
- *Fat content*	150 (17.6%)	46 (23.0%)	9 (4.5%)	6 (3.0%)	78 (55.7%)	11 (10.0%)
- *Sodium content*	150 (17.6%)	46 (23.0%)	9 (4.5%)	6 (3.0%)	78 (55.7%)	11 (10.0%)
**Total number of products presenting a nutrition content claim**	86 (10.1%)	27 (13.5%)	8 (4.0%)	0 (0.0%)	43 (30.7%)	8 (7.3%)
- *Low energy claim*	41 (4.8%)	7 (3.5%)	8 (4.0%)	0 (0.0%)	25 (17.9%)	1 (0.9%)
- *Low carbohidrate claim*	50 (5.9%)	23 (11.5%)	0 (0.0%)	0 (0.0%)	23 (16.4%)	4 (3.6%)
- *Low sugar claim*	50 (5.9%)	10 (5.0%)	0 (0.0%)	0 (0.0%)	32 (22.9%)	8 (7.3%)

**Table 4 ijerph-19-04609-t004:** Proportion of alcoholic beverages that present nutrition-related information, of those manufactured by manufacturers with five or more beverages included in the study.

Alcoholic Beverage Manufacturer	Percentage of Assessed Beverages That Presented Nutrition-Related Information (%)
Bundaberg Distilling Company	9/9 (100.0%)
Smirnoff	11/12 (91.7%)
United Distillers Limited	8/8 (100.0%)
Ampersand Projects	5/6 (83.3%)
Gordon’s	5/6 (83.3%)
Coopers Brewery	2/5 (40.0%)
Orlando Wines	2/5 (40.0%)
Rekorderlig	2/6 (33.3%)
Carlsberg Group	3/10 (30.0%)
Scape Goat	1/5 (20.0%)
Strongbow	1/5 (20.0%)
Asahi Beverages	3/17 (17.6%)
Jack Daniel’s	1/6 (16.7%)
Koppabergs Brewery	0/7 (0.0%)
Little Fat Lamb	0/8 (0.0%)
The Hills Cider Company	0/5 (0.0%)
Wild Turkey	0/5 (0.0%)

## Data Availability

The data presented in this study are available on request from the corresponding author. The data are not publicly available due to technical constraints.

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
