# Peer review of "Nutrition-Related Information on Alcoholic Beverages in Victoria, Australia, 2021"

_ijerph, 2022, doi:10.3390/ijerph19084609_

Round 1

Reviewer 1 Report

Few studies have examined the prevalence of nutrition information on alcohol containers, and none have explored the format in which nutrition information is presented on alcohol. The aim of this study was to investigate the prevalence and format of nutrition-related information on packaged alcoholic beverages in Australia using in an in-store audit in July 2021. This paper is well written and explores a novel topic by leveraging knowledge and expertise of food labelling experts and applying to alcohol. Results, although limited in generalizability beyond Victoria, will make a contribution to the literature. I enjoyed reading this paper and have made suggestions below for the authors to consider.

Lines 58-60: Include examples of jurisdictions that with slightly different legislation regulating nutrient claims on alcohol to highlight potential policy options. For example, Canada permits nutrient content claims only (e.g., 0 g sugars) but not nutrient content statements (e.g., sugar free). Also, any container presenting nutrient content claims must provide a NIP.

Canadian Food Inspection Agency. Industry Labelling Tool. Nutrition labelling - Reasons for losing the exemption [Internet]. Ottawa, ON; 2021 May. Available from: https://inspection.canada.ca/food-label-requirements/labelling/industry/nutrition-labelling/exemptions/eng/1389198015395/1389198098450?chap=0#c3

Lines 67-70: Include a comment about the Australian government exploring options for providing energy information on alcohol container, as well as the various formats for presenting this information.

https://www.foodstandards.gov.au/consumer/labelling/Pages/Energy-labelling-of-alcoholic-beverages.aspx

Materials and Methods

Line 110: What is a “Victorian” store? Is this a story in the state of Victoria?

Line 134: What is meant by “…sample included beverages from most or all shelves of beverages…”. Why would only most shelves and not all shelves of beverages being included in the sample?

Line 135: Given that the sampling continued until the target sample size was achieved, did the starting point within the retailer influence the sample? For example, in Canada, liquor retailers organize beverages by place of origin, craft vs commercial brands, subcategory of alcohol (e.g., red wine, white wine, brown rum, white rum).

Were you able to capture if a container included a nutrient level on the top or bottom ribbon versus in the centre of the principal panel of the container? In any instances was nutrition information positioned on the side of the container vs front/back of container?

Results

I had to reread the Introduction and Results sections to understand differences in nutrient levels, nutrient content information, nutrition content claim, nutrition statement, etc.

Could you please provide more details in the methods about the definitions of key variables to clarify to an international audience what is being captured on the container and in which format? For example, in Table 2:

  • Is “nutrient level” the nutrient content in numeric form (e.g., 0 g sugars, or a NIP)?
  • I am not clear how 9 wine containers presented a NIP only but 8 presented both nutrient levels and a nutrition content claim? What does that look like? What is a statement on the back of a beverage?
  • Do the results suggest some alcohol containers present nutrition content claims but no NIP, which is in violation of the regulations in Australia?

Also, to clarify, the Australian regulations do not require a NIP when nutrient content information (e.g., 90 calories) are presented on the front-of-pack label of the container?

In Table 3, would any container that provides a full NIP be counted as providing the full list of energy and nutrient levels?

Lines 197 – 198: I believe words are missing: “The prevalence of nutrient content claims related to…sugars content, were observed in similar proportions”?

Did the alcoholic beverages that did or did not display nutrition information differ in alcohol strength?

Discussion

Line 218-220: What is the limitation of permitting manufacturers to provide nutrition information on select products versus across alcohol products? Also, if an alcohol container presents “0 g sugars” on the label, this is technically not an interpretative label but can only be interpreted as no sugars. Given that the nutrition information is only on select products, do consumers interpret the nutrition information positively?

Lines 247-259: This is a nice discussion of potential alcohol control policies that can be considered. A strong body of literature demonstrates the effectiveness of alcohol price and availability policies for reducing per capita alcohol use and harms. However, what is unique about product container or package label interventions in influencing consumer purchase and use decisions compared to other types of information-based interventions (e.g., in-store signage). What role can policies controlling alcohol container label information play in a more comprehensive alcohol strategy?  

Lines 249-253: Please cite recently published review examining the effectiveness of alcohol container labels with health warnings for increasing consumer knowledge of alcohol-caused health harms and reducing per capita alcohol use.

Kokole D, Anderson P, Jane-Llopis E. Nature and potential impact of alcohol health warning labels: a scoping review. Nutrients 2021, 13. https://doi.org/10.3390/nu13903065

Conclusions

Lines 287-291: Is this meant to be 2 separate sentences? Please review and revise as it is unclear.

Reviewer 2 Report

The manuscript is readable ad covers an interesting topic.

The analysis is far from sophisticated but seems appropriate given the intent of the authors. Due consideration was given to the issue of inflated Type I error from multiple testing.

The systematic sampling procedure makes sense in context. A random start is really important prior to picking every other item, and I don't believe the authors were clear about whether such was the case.

The conclusion provides some implications of the study. It would be helpful for the authors to flesh out somewhat more concrete opportunities for lessons learned that can be applied to subsequent policy and practice.

Reviewer 3 Report

A very interesting job. However, I believe that the authors should present the research material in more detail. I also miss an in-depth statistical analysis and more precise conclusions. Authors should also write about the limitations of the work.

Reviewer 4 Report

This study aimed to investigate the prevalence of nutrition-related information on packaged alcoholic beverages in Australia. Although the study is well-planned, the rationale and the discussion can be improved. For instance, the authors mentioned the 'health halo' effect, but it is not clear that this can be a case for alcohol drinkers. Additionally, alcohol drinkers have different characteristics than the general population, so their perception about the healthiness of alcohol or any other products might not be the same as non-drinkers who are more conscious about what they eat or drink. This issue could be discussed or given in the introduction section. 

I suggest authors sub-group the methods section. Are ready to drink alcoholic beverages same with alcopops? If there are any, it would be better to explain the differences between nutrition content claim and nutrition claim. 

Round 2

Reviewer 4 Report

I would like to thank the authors for addressing my comments.